# Pupillary Responses to Faces Are Modulated by Familiarity and Rewarding Context

**DOI:** 10.3390/brainsci11060794

**Published:** 2021-06-16

**Authors:** Magdalena Matyjek, Mareike Bayer, Isabel Dziobek

**Affiliations:** 1Berlin School of Mind and Brain, Humboldt-Universität zu Berlin, Luisenstr. 56, 10117 Berlin, Germany; mareike.bayer@hu-berlin.de (M.B.); isabel.dziobek@hu-berlin.de (I.D.); 2Institute of Psychology, Humboldt-Universität zu Berlin, Rudower Chaussee, 12489 Berlin, Germany

**Keywords:** social reward, familiarity, social relevance, pupil size, feedback

## Abstract

Observing familiar (known, recognisable) and socially relevant (personally important) faces elicits activation in the brain’s reward circuit. Although smiling faces are often used as social rewards in research, it is firstly unclear whether familiarity and social relevance modulate the processing of faces differently, and secondly whether this processing depends on the feedback context, i.e., if it is different when smiles are delivered depending on performance or in the absence of any action (passive viewing). In this preregistered study, we compared pupillary responses to smiling faces differing in subjective familiarity and social relevance. They were displayed in a passive viewing task and in an active task (a speeded visual short-term memory task). The pupils were affected only in the active task and only by subjective familiarity. Contrary to expectations, smaller dilations were observed in response to more familiar faces. Behavioural ratings supported the superior rewarding context of the active task, with higher reward ratings for the game than the passive task. This study offers two major insights. Firstly, familiarity plays a role in the processing of social rewards, as known and unknown faces influence the autonomic responses differently. Secondly, the feedback context is crucial in reward research as positive stimuli are rewarding when they are dependent on performance.

## 1. Introduction

### 1.1. Privileged Processing of Familiar Faces in Humans

Familiarity of faces is an important factor in the socio-cognitive functioning of humans. Immediate access to information about familiar others is crucial for successful social interactions [1]. Indeed, numerous studies have shown evidence of preferential processing of familiar faces. The familiarity of a face dramatically facilitates its recognition [2,3], enhances the cueing effects of the eyes [4], and also requires less attentional resources and no conscious awareness [5]. Social cues conveyed by familiar faces are processed and recognised faster than cues from unfamiliar faces [6], and on the emotional level, familiarity facilitates one’s empathy towards the other [7]. Taken together, it seems that familiarity plays an exceptional role in human socio-cognitive functioning, which is furthered by research on social impairments, such as autism spectrum conditions (ASC). Abnormal patterns of neural activation have been reported in this population in response to unfamiliar faces (for a review, see [8]). These include both hypoactivation of individual cortical areas such as the fusiform gyrus, the amygdala, and the superior sulcus, and the atypicalities of distributed cortical and subcortical brain networks. However, there is evidence for typical patterns of neural activation (mainly in the fusiform gyrus and the amygdala) for familiar faces in this group [9,10]. This suggests that familiarity may normalise otherwise aberrant neural responses to faces in individuals with ASC. Altogether, familiarity has a unique standing in the human social functioning, which has even led to a shift in the proposition of humans being face experts, to that of humans being familiar face experts [11].

### 1.2. Familiarity and Social Relevance of Faces

Although it is clear that the faces of familiar persons are processed differently than faces of strangers, it is important to note that there are different types of familiar faces we encounter in life: relatives, friends, colleagues, celebrities, etc. [12]. Among the qualitative differences between these categories, social relevance is the most notable. Close relatives and loved ones are more socially relevant than co-workers, and a superior is more socially relevant than an acquaintance from the gym. For the purpose of operationalising the key terms in this article, we propose the following definitions: Familiar persons are those that one recognises and has some knowledge about, but who are not necessarily personally important for the person (e.g., actors, a frequently seen salesperson in a grocery store); Socially relevant persons are those connected to one on a personal level, sharing a social context of special subjective meaning, and bearing personal importance (in the literature, social relevance is also referred to in a broader framework as ‘personal relevance’ [13] or ‘personal importance’ [14]), e.g., friends, relatives, school teachers. Both familiarity and social relevance describe spectra rather than binary categories as faces can be more (or less) familiar and more (or less) socially relevant. Moreover, socially relevant faces are often linked to affective knowledge and emotional responses of varying intensity, ranging from love for relatives and partners to feelings of acceptance or belonging in a new group context.

Following these definitions, all socially relevant faces are necessarily familiar, but not all familiar faces are socially relevant. Importantly, relevance is not merely a feature of higher levels of familiarity. For example, the face of a news presenter may be very familiar but may still not bear any social relevance due to a lack of personal importance to the person. These features can be processed detachedly, which is supported by studies with patients suffering from Capgras delusion, who believe that their loved ones are in fact imposters. Such patients can typically recognise a familiar face, but they show no autonomic response (skin conductance response) that is typically observed when seeing loved individuals [15]. Thus, the recognition process (crucial for familiarity) and the autonomic nervous system response (linked to highly socially relevant individuals) are independent to a degree. However, few studies have compared the impact of familiarity and social relevance on the processing of faces. In one such study, the N170 component was shown to be increased for personally important faces (i.e., those of a participant’s mother and their own) in contrast to less familiar (celebrities) or unknown faces [14]. In another study, familiarity (friend and romantic partner vs. stranger) influenced neural processing earlier than love, i.e., high personal relevance (romantic partner vs. friend and stranger; [13]). It is important to note that the definitions of familiarity and social relevance vary between studies and operationalisation of those terms is crucial for between-study comparisons.

### 1.3. Social Relevance and Reward Circuitry

Importantly, a number of neuroimaging studies have reported that faces of beloved individuals (i.e., familiar and socially relevant persons for whom one has strong positive emotional feelings) elicit stronger responses of the reward circuitry (among others the ventral tegmental area, striatum, anterior and posterior cingulate) than faces of less familiar persons [13,16,17,18,19]. Thus, observing emotionally associated and socially relevant faces is more rewarding than observing faces of less familiar and relevant individuals or strangers. However, in reward research, where smiling faces are often used as social rewards, most studies use faces of individuals unknown to participants (not familiar and not relevant). This is surprising in light of the neuroimaging studies showing that the reward circuit is uniquely activated by highly relevant faces [15]. Hence, it is important to empirically address whether increasing levels of familiarity and/or social relevance are linked to other reward responses than the neural responses, including behavioural self-reported reward values and psychophysiological indexes.

### 1.4. Reward as Property of a Pleasant Stimulus vs. as Outcome Contingent on Behaviour

It is important to note that the reward magnitude of a positive social stimulus is substantially different when its presence is contingent on one’s behaviour in contrast to when it is passively viewed regardless of one’s behaviour. For example, a mother’s smile in response to a child’s appropriate social behaviour differs from a smiling face in a commercial viewed on a television screen (for a discussion, see [20]). A reward value is not merely a property of a stimulus, but also lies in the receiver’s subjective judgement of the stimulus [21]. However, little research has explored the effect of familiarity (and social relevance) on the rewarding and motivating value of faces serving as feedback contingent on behaviour. Two studies investigated reward responsiveness to familiar and unfamiliar faces in children with and without ASC and found effects of the condition [22,23]. One explanation for this surprising result in light of the known preferential processing of familiar faces in ASC [10] might lie in the lack of contextual importance of the faces in the study. It is possible that for autistic individuals, the reward value of the faces serving as feedback for recent actions depends on the believability that this person offers such feedback in the given situation. Thus, it is not the smiling relevant face per se that is rewarding, but rather the smile of a relevant person responding to one’s actions. An argument for such disentanglement of face processing and context-dependent social meaning in individuals with ASC is offered by an observation of simultaneous normal activation in the fusiform gyrus and reduced activity in the cingulate cortex in response to specifically familiar faces [10]. The fusiform gyrus is linked to face processing and the cingulate cortex is a part of the ‘default network’ responsible for mentalising and social processing. This could suggest that the interplay of familiarity/social relevance and reward is dependent on the social context.

### 1.5. Familiarity and Social Relevance in Active and Passive Tasks

Overall, the literature suggests that faces of familiar and socially relevant persons trigger activation in the brain structures devoted to the processing of rewards. However, more research is needed to learn whether their rewarding value (1) increases with the faces’ increasing familiarity and/or social relevance and (2) depends on behaviour contingency (i.e., that one has to work towards getting them or just view them passively).

In this study, we investigated whether the reward value of smiling faces depends on a feedback context and/or familiarity and social relevance. To this end, we included two tasks, namely an active task, in which participants played a repeat-a-pattern game and received social rewards (photos of smiling faces) on successful trials only (reward contingent on behaviour); and a passive task, in which participants viewed the same smiling faces with no context of performance-feedback (non-contingent presentation of the face).

We aimed to create a set of pictures of smiling faces that would ensure a wide range of levels of familiarity and social relevance for each individual, but similar levels across all participants. For this, in the present study, participants observed smiling faces of three types: (1) strangers, who are unfamiliar and socially non-relevant persons; (2) celebrities, who are personally non-relevant persons with familiar faces; and (3) experimenters, who become familiar and, to some extent, socially relevant through the importance of the shared social context of the experiment. In order to ensure the familiarity and social relevance of the experimenters, a scripted social interaction was introduced during the course of the participants’ study appointment in the lab that entailed, among others, semi-scripted conversations (for similar designs, see [24,25]). To capture expected intraindividual differences in the perceived familiarity and relevance of the faces, and to reflect them in the data, participants provided subjective ratings of the depicted persons. We then used the ratings as predictors of physiological responses to these faces.

### 1.6. Measuring Reward Processing with Eye-Tracking Pupillometry

While most reward processing studies targeting responses to familiar vs. unfamiliar faces exploit neuroimaging methods, a worthy alternative is offered by measures of the central autonomic nervous system. Pupillary responses measured with eye tracking technology allow researchers to grasp a full picture of its responsiveness, i.e., they capture the influence of both the sympathetic and the parasympathetic branches. This includes dilations caused by excitation of the sympathetic or inhibition of the parasympathetic branches and constrictions caused by the excitation of the parasympathetic or inhibition of the sympathetic branches [26]. A further advantage of this technology is its non-invasiveness, which adds to the naturalness of the social context in a laboratory setting. The pupil systematically dilates in response to mental processes, such as cognitive activity, mental effort, or increasing levels of arousal [27]. Importantly, the increase in its size has been linked to goal-priming with rewards [28], to higher magnitude of possible rewards [29], and to reward anticipation [30,31]. Similarly, pleasant (and rewarding) images of smoking-related cues trigger an increase in smokers’ pupil sizes [32]. Moreover, pupillary responses have previously been used to discriminate between familiar and unfamiliar faces [33]. Finally, pupil size strongly correlates with the activation of the locus coeruleus (LC; [34]), which plays an important role in reward processing and motivation [35]. The size of a pupil is thus a promising indicator of the subjective reward value of an observed stimulus.

### 1.7. Aims and Hypotheses

The aim of the current study was to investigate pupillary responses (as indicators of reward processing) to smiling faces varying in their subjective levels of familiarity and social relevance. We hypothesised that the familiarity and social relevance of smiling faces, as measured via subjective ratings, would be linked to increased pupil sizes, especially in the game. This is based on: (1) the assumption that feedback from relevant and familiar faces would be more rewarding than from unknown and irrelevant faces, and (2) previous research showing that more familiar persons are regarded as being more arousing [36] and that changes in pupil sizes can be linked to arousal as an indicator of the motivational and rewarding features of stimuli (e.g., [29]). We did not have a directed hypothesis as to which factor, familiarity or social relevance, would influence the pupillary responses more. We also aimed to analyse reaction times to check whether familiarity and social relevance influence performance in the following trial, as they have previously been shown to differentiate between reward conditions in speeded tasks [37]. Again, we expected that more familiar and relevant faces would improve the subsequent performance (i.e., shorten reaction times).

We also performed two secondary analyses. First, we measured participants’ autistic traits to investigate their possible modulatory effects on pupil responses to smiling faces in this study. The reward value of social stimuli such as faces is proposed to be reduced in individuals with autism [38], and studies that measured pupil size have indicated abnormal patterns of dilation in this population [33,39]. For this secondary analysis, we predicted that less familiar and less relevant faces would elicit smaller dilations in individuals with higher levels of autistic traits. The reasoning is that even though social stimuli may have lower reward value for individuals with higher levels of autistic traits [40], the familiarity (and social relevance) of faces likely normalises their processing (as is the case in ASC; [10]).

Finally, since the attractiveness of observed faces has previously been shown to influence pupil sizes [41], we collected subjective ratings of attractiveness of the faces presented in this study and included them as a covariate (predictor of no major interest) in the analyses. We also aimed to explore the correlation between ratings of attractiveness and reward value. A positive correlation would offer support for considering attractive faces as being akin to rewards [42].

## 2. Materials and Methods

Methods, power analyses and hypotheses were preregistered on the 25th of November 2019 at https://www.osf.io/h4awf (accessed on 16 June 2021). The data and analysis code in R for the current study (as well as an html file presenting all the analyses in an accessible way without the need to run the code) are available in the OSF repository: https://osf.io/623jg/ (accessed on 16 June 2021).

### 2.1. Sample Determination and Participants

Prior to data collection, a power analysis was performed with G*power software [43] for fixed effects in linear multiple regression, with power set to 0.8, alpha set to 0.05, two predictors (familiarity and relevance), and the total number of included predictors set to four (familiarity, relevance, attractiveness, and trial number). This analysis showed that, to observe a medium effect size of f^2^ = 0.15, a total sample size of 68 is required. Participants were recruited via eBay (a popular online advertising service in Germany), social media, flyers distributed on the university’s campus, and through participant databases of the Berlin School of Mind and Brain and Humboldt-Universität zu Berlin. All participants were between 18 and 40 years of age, had no self-reported history of psychological illness in the last six months, were proficient in English, and had normal or corrected-to-normal vision. A total of 84 volunteers participated in the study. Fourteen data sets were subsequently rejected due to poor data quality or failed attention checks (for details, see Section 2.6). The remaining sample of 70 participants consisted of 45 females and 25 males, with an average age of 27.77 years (*SD* = 5.17), which did not differ between the genders, *t*(44.74) = −0.45, *p* = 0.66. The study was approved by the Ethics Committee of the Institute of Psychology, Humboldt-Universität zu Berlin (nr 2019–24). Participants provided prior informed written consent.

### 2.2. Stimuli and Materials

The stimuli set consisted of pictures of 10 females: two experimenters, two strangers, and six celebrities. All experimenters and strangers consented in writing to the use of their photographs in the study. Pictures of celebrities were selected from the internet. We targeted pictures of popular actresses and singers aged between 20 to 40 years without excessive make up, with a straight gaze and smile, who were facing the camera. The pictures of experimenters and strangers (personal contacts of the researchers with no connection to the study) were taken according to these criteria. The background was removed from all photos, which were resized to 238 × 238 pixels (seven visual angles) and transformed into greyscale. Finally, their brightness and contrast were adjusted so that all matched in terms of luminance (ensured with the mean value of luminance in perceptual space in GIMP 2.0, which was additionally confirmed with a photometer). Photo editing was conducted with GIMP 2.0.

The experiment was run using a 19-inch flat-screen monitor with 1024 × 1280 pixel resolution and a 60 Hz refresh rate. The experiment was programmed and executed in MATLAB. Pupillary responses were recorded binocularly using a desktop-mounted eye tracker (Eye Tribe, TheEyeTribe) at a 60 Hz sampling rate and the EyeTribe Toolbox for MATLAB [44]. Eye Tribe provides pupil measurements in arbitrary units (not mm or pixels). Prior to each task, the eye tracker was calibrated with a nine-point grid. Calibration was accepted when <0.7 degree of accuracy was achieved.

### 2.3. Procedure and Tasks

#### 2.3.1. Socialising

Upon arrival, participants were fetched at the entrance of the university building by one of two experimenters (randomly assigned), who introduced herself and maintained a semi-scripted, naturally flowing social conversation on their way to the laboratory. There, the experimenter explained the study and her role in it (a master student doing a lab rotation) and encouraged questions. After acquiring signed consent, she presented the participants with all the face stimuli and asked them to indicate the faces they recognised. This was done to ensure that participants recognised their experimenter in the picture, recognised at least one celebrity (and did not exhibit any excessive positive or negative affection towards them), and were unfamiliar with at least one stranger. Based on their answers, the experimenter selected one celebrity and one stranger. This part also served as further natural socialising of the participants and the experimenter, who led a light, semi-scripted social conversation to increase their social relevance (shared context and social interaction) and familiarity through an extended exposure (approximately 10-min conversation with various expressions and viewing angles of the experimenter’s face). The overall duration of the interaction between experimenters and participants prior to the tasks was approximately 20 min.

#### 2.3.2. Tasks

The lab was an artificially lit room (with constant illumination for all participants) with covered windows to keep the light conditions constant. A chinrest was used to limit head movements and to maintain the distance between participants and the screen at 50 cm. All participants completed two tasks in counterbalanced order: a passive viewing task and a repeat-a-pattern game. Prior to each, the eye tracker was calibrated. Participants were instructed to keep their eyes fixed on the centre of the screen.

In the repeat-a-pattern game modelled on the popular Simon game (Hasbro Gaming), participants were instructed to quickly repeat a pattern of appearance of four coloured dots that was presented on the screen (in terms of locations and colours), by pressing the corresponding buttons on a gamepad (both in terms of locations and colours, i.e., replicating the pattern). They were informed that they would see a smiling face as positive feedback in case of success, or a red cross in case of failure. To further their motivation in the game through an element of competition, participants could also place their nicknames and scores on the wall of best scores after completing the tasks. The game consisted of six blocks of 18 trials, resulting in a total of 108 trials (circa 8 min). In each trial, participants first saw a fixation cross for 500 ms. Then, four circles (50 pixels diameter) of red, yellow, green, or blue colour were presented with 0.1 s blank screen in between. The circles were displayed 50 pixels from the centre of the screen reflecting the topography and colours of the gamepad’s buttons. To create a dynamic setting and sustain motivation to play, in the first two blocks the display time of the circles was 0.2 s, in the next two blocks 0.17 s, and in the last two blocks 0.14 s. After the pattern was shown, the word ‘GO!’ was displayed in the centre of the screen for 0.2 s, triggering the response. Participants then repeated the pattern on the gamepad, which was followed by a 0.5 s blank screen and feedback. In case they failed to press four buttons within 3 s or to repeat the pattern correctly, the trial was unsuccessful, in which case a red cross was presented in the centre of the screen for 0.5 s. In successful trials, a smiling face of a stranger, celebrity, or the experimenter (in a randomised order) was presented at the centre of the screen for 3 s.

In the passive viewing task, the same pictures (of the same experimenter, celebrity, and stranger as in the game) were displayed in random order for 3 s with a 500 ms inter-trial interval. As in the game, the 108 trials were divided into six blocks (circa 6 min). Additionally, to ensure that participants paid attention to the faces, a 1-back task was introduced: Eight times throughout the task (at least once in a block) one of the faces was presented in a red frame with a question ‘is this the face you saw in the last trial?’; participants responded yes or no by pressing a button. Since the face had to be stored for only 0.5 s (the intertrial interval), attentional and memory demands were likely very low.

### 2.4. Ratings and Questionnaires

After completing both tasks, participants rated the faces used in the tasks on the dimensions of social relevance, familiarity, attractiveness, and reward value, via an online survey administered with the SoSci Survey platform, www.soscisurvey.de [45]. Participants answered the following questions (separately for each factor): “How socially relevant/familiar/attractive do you find the people presented in the pictures at the moment?/How rewarding did you find these pictures?” Additionally, short descriptions were provided to ensure all participants shared an understanding of the concepts in question. These were, respectively to the questions: “a socially relevant person is someone you are connected to, who is important for you for some reason, who you share a social context with”; “a familiar person is someone you know, recognise, have some knowledge about”; “an attractive person is someone you find aesthetically pleasant, pleasing, interesting, beautiful, arousing, or desirable”; “when you saw these faces as feedback in the game with the coloured buttons/the passive looking task, how rewarding were they to you?” We were aware of the fact that the reward ratings for faces in the passive viewing task were not reflecting rewards per se, as in this task the smiling faces were positive stimuli presented in the absence of any actions performed by the participants. However, to allow exploratory comparison between tasks, we asked for the reward ratings of the faces in both tasks (repeat-a-pattern game and the passive viewing). All ratings were given on a 100-point scale with extremes labelled as ‘not at all’ (0) and ‘very’ (100).

For an analysis of autistic traits’ effect on arousal measurements in the tasks, participants also filled out the 10-item Autism Spectrum Quotient (AQ; [46]). Additionally, they filled out the behavioural inhibition/approach scales (BIS/BAS; [47]), which addresses whether individuals are motivated by pursuit of rewards (the BAS system) or avoidance of punishment (the BIS system), and the Liebowitz Social Anxiety Scale (LSAS; [48]), which addresses anxiety linked to social stimuli and situations, e.g., social judgement.

### 2.5. Data Preprocessing

#### 2.5.1. Pupillary Responses

Offline pre-processing of the pupillary data was performed using MATLAB, with a procedure proposed by Kret & Sjak-Shie [49] with their default settings. Pre-processing included blink and missing data interpolation (artifacts identified as dilation speed outliers and edge artifacts, trend-line deviation outliers, and temporally isolated samples), filtering, calculating mean pupil size from both eyes, smoothing, and up-sampling to increase the temporal resolution and smoothness (for details, see [49]). Segmentation and subtractive 200 ms baseline correction were subsequently performed in R ver. 4.0.2 [50]. The mean pupil size across a time window within the segment was calculated and used in the analyses. The time window was based on visual inspection of the averaged pupillary responses (see Figure 1) resulting in 1–3 s for the game and 0.5–2 s for the passive task. The difference in the time windows (which were used for response averaging) results from the different overall shape of the phasic pupil responses to stimuli presentations in the repeat-a-pattern game and the passive viewing (for details, see Section 3.3). Averaging of trials was performed to increase the signal-to-noise ratio of the signal.

#### 2.5.2. Reaction Times in the Game

Reaction times were recorded as the last button press in an attempt to repeat the pattern in the game. Unsuccessful trials (incorrect responses or responses that took longer than 3 s) were removed from the data sets (16.44 trials on average). Three participants showed low accuracy, i.e., lower than 2 SD from the sample’s mean (on average, participants were successful in 92 trials). We built models both with and without the data from these three participants and observed similar results. We therefore here describe the models generated without the exclusion of these participants, although for the sake of completeness, both models can be found in the analysis code (https://osf.io/623jg/ (accessed on 16 June 2021), Section 5.1.7). Trials with reactions times longer or shorter than 2 SD from the mean of each participant and longer than 3 s were removed from the data set (4.11 trials on average).

### 2.6. Data Rejection

As registered, full data sets were removed if participants failed to give the correct response in four or more trials of the 1-back task in the passive viewing task (two data sets; for details, see Section 2.3.2), or if the data quality of the acquired signal was insufficient. Insufficient data quality was defined as 50 or more percent of trials rejected due to missing data samples within each trial (50% or more) or more than 50% missing data samples in a trial’s baseline. Missing data samples were mainly blinks and fixations away from the stimuli. Since rotating the eyeball to look at distant points from the centre of the screen causes an artificial decrease of the pupil size [49], all fixations beyond the area of interest were removed. The area of interest was defined as the size of the stimuli plus two visual angles (not one as preregistered, to match the previous literature, [51]).

These restrictions led to the rejection of 14 data sets. In the remaining data from 70 subjects, the number of trials across face types (experimenter, stranger, celebrity) was not significantly different in either of the two tasks, *F*(2,207) = 0.009, *p* = 0.99 (game) and *F*(2,207) = 0, *p* = 1 (passive task).

### 2.7. Data Analyses

All data analyses were performed using R ver. 4.0.2 [50]. The significance level for all the tests was set to 0.05. The analysis of the game data only included successful trials (where faces were presented as positive feedback). For all analyses (subjective ratings, pupillary responses, reaction times), we used multiple regression analyses with mixed effects with the lmerTest package ver. 3.1-2 [52] and with treatment contrasts. Random intercepts for participants were used in all models. Additional random intercepts for stimuli (10 pictures used in the study) were used in the models, but in the game data (both pupillary responses and reaction times) variance of this term was 0, which suggests a singular model, and thus this term was eliminated from the game model. Assumptions for multiple regression were checked (normality, linearity, multicollinearity, homoscedasticity). The social relevance and attractiveness ratings models showed moderate skewness in the residual plots and were subsequently re-fitted with transformed data. The distribution of residuals in the initial reaction times model was positively skewed, which violated the normality assumption for linear models. Instead of transforming the data, which would make the estimates not readily interpretable [53], we used a generalised linear mixed model with raw reaction times with fitted inverse Gaussian distribution and the identity link function, which also ensured that the estimates could be considered direct effect sizes. Confidence intervals for this analysis were calculated via the confint function of the R base package stats with method Wald. Marginal and conditional R^2^ were calculated as measures of goodness of fit for mixed models [54], in which marginal R^2^ (R^2^_m_) reflects variance explained by fixed factors, and conditional R^2^ (R^2^_c_) the variance explained by the entire model. Since there is no agreement on a method for estimating standardised effect sizes for individual terms in linear mixed effect models [55], we used an indirect method for their estimation: partial Cohen’s fs (*f_p_*) were calculated with effect size package v. 0.3.2 [56], which calculates these from an analysis of variance run on the models. The p-values were computed via Wald-statistics approximation (treating t as Wald z), and corrected with package multcomp v. 1.4-13 where appropriate [57]. *p*-values for exploratory analyses are intentionally not provided. Plots and tables of the models were created with the sjPlot package v. 2.8.4 [58].

## 3. Results

### 3.1. Subjective Ratings—Planned Analyses

Table 1 shows the mean ratings (with standard deviations) and results of analysis of variance (ANOVA) performed on regression models with random intercepts for subjects and (1) one predictor: face type (stranger, experimenter, celebrity) for social relevance, familiarity, and attractiveness, or (2) two predictors: face type and task (repeat-a-pattern game, passive task) and their interaction for reward value. Once again, the analyses including reward value of faces in the passive task were included for completeness, but the meaning of reward value across the two tasks is likely different due to the presence (in the repeat-a-pattern game) or absence (in the passive viewing) of the feedback (and thus reward) context.

Face type was a significant predictor in all models with one predictor. Post hoc analyses (with Holm correction) revealed that, in terms of social relevance, the experimenter was rated significantly higher than both the celebrity and the stranger, and the celebrity was also rated higher than the stranger (all *ps*_corr_ < 0.001). Both the experimenter and the celebrity were rated as more familiar than the stranger (*ps*_corr_ < 0.001), but similarly to each other (*p*_corr_ = 0.13). On average, the celebrity was rated as being more attractive than the experimenter and the stranger (both *ps*_corr_ < 0.001). Altogether, these results suggest that the experimenter and the celebrity were perceived by the participants as being more familiar and socially relevant than the stranger. The experimenter was also perceived as being more socially relevant, but not more familiar, than the celebrity.

Exploratory analysis of the reward value of faces revealed an interaction of face type and task, which is presented in Figure 2a. Post hoc comparisons (the complete list of adjusted *p*-values can be found in the Appendix A) confirmed that in the game, the experimenter’s face was perceived as more rewarding than the other two faces, whereas in the passive task the most rewarding face was that of the stranger. Moreover, while both the experimenter and the celebrity were rated as more rewarding in the game than in the passive task, it was the opposite for the stranger.

### 3.2. Subjective Ratings—Exploratory Analysis

We explored correlations between the subjective ratings. We were particularly interested in the relationship between attractiveness and reward value. A correlogram with all ratings, shown in Figure 2b, suggests that higher perceived attractiveness is linked to higher reward value, but only in the repeat-a-pattern game.

### 3.3. Pupillary Responses—Planned Analyses

The average pupil sizes in response to the stimuli (stranger, celebrity, and experimenter) in both tasks are shown in Figure 1. Importantly, these categories were not used as predictors in the regression models; instead, we used the subjective ratings provided by the participants. The overall shape of the pupil responses reflected the differences between the tasks. In the repeat-a-pattern game, stimuli were presented as feedback to the participants’ responses, which caused large pupil dilations, reflecting the cognitive load of the task (see Appendix A). In turn, this caused the pupils to decrease in size during the feedback (face) presentation (return to baseline). Since in the passive viewing task there were no systematic large dilations prior to the stimulus onset, the pupil responses in this task showed less fluctuation.

For both tasks, we built multiple regression models with continuous and centred subjective ratings of social relevance and familiarity of the stimuli as main predictors. The ratings of reward value were also included in the game model, but not in the passive viewing model (as reward value in the context of passive viewing is not meaningful). We controlled for attractiveness and time effects by including attractiveness ratings and trial numbers as covariates. The models are shown in Table 2.

In the game, we observed a statistically significant effect of familiarity, with smaller pupil sizes for more familiar faces. The results also show the effects of trial number in both tasks such that, in the repeat-a-pattern game, the pupil sizes increased with time, while in the passive viewing task they decreased with time. In the repeat-a-pattern game, the effect sizes were: *f_p_* = 0.03 for trial, *f_p_* = 0.03 for familiarity, all others ≤ 0.02; in the passive task, the effect sizes were: *f_p_* = 0.07 for trial, all others ≤ 0.05.

We tested gender, LSAS, and BIS/BAS scales to check whether they would improve the models’ fit. As none of these aspects did, they were thus not further considered. Furthermore, we tested whether the two experimenters in the study had different effects on the pupillary responses and found no such effect (for both analyses, see points 4.1.5 and 5.1.5 in the analysis code).

### 3.4. Reaction Times—Planned Analysis

Since in the repeat-a-pattern game participants did not know which face they would see after a successful repetition of a pattern in each trial (face type was randomised for all participants), we built a generalised linear mixed model predicting reaction time in the subsequent trials, as a possible indicator of increased performance following a rewarding stimulus. This analysis yielded a large effect of trial, *est*. = −2.26, *95% CI* = −2.40–−2.11, *t* = −31.14, *p* < 0.001, showing that, on average, participants improved their reaction times in each trial by 2.26 ms. No other rating significantly predicted reaction times: familiarity (*est*. = −0.03, *95% CI* = −0.25–0.18, *t* = −0.32, *p* = 0.75), social relevance (*est*. = −0.02, *95% CI* = −0.23–0.20, *t* = −0.15, *p* = 0.88), attractiveness (*est*. = 0.13, *95% CI* = −0.11–0.36, *t* = 1.07, *p* = 0.29), reward value (*est*. = 0.06, *95% CI* = −0.14–0.26, *t* = 0.60, *p* = 0.55). The overall fit of the model was R^2^_m_ = 0.37 and R^2^_c_ = 1.

### 3.5. Pupillary Responses—Exploratory Analysis with Autistic Traits

We built additional models for each task including the AQ scores and their interactions with the stimuli’s familiarity and social relevance to explore their possible effects on the pupillary responses. The models are presented in Table 3.

Although autistic traits (AQ score) were not a promising predictor of pupil size in either of the tasks (game: *f_p_* = 0.05, passive task: *f_p_* = 0.02), the confidence intervals and the standard error for the estimates in the passive viewing task suggest that a noteworthy amount of variance was explained by the AQ × familiarity interaction (*f_p_* = 0.04). A visual inspection of this interaction (Figure 3) showed numerically larger pupil sizes for higher autistic traits in high familiarity and smaller pupil sizes in increased autistic traits in low familiarity.

## 4. Discussion

In this study, we explored pupillary responses to smiling faces differing in their subjective familiarity, social relevance, and reward value. We considered these faces as social rewards either contingent on behaviour (in a repeat-a-pattern game) or not (in a passive viewing task). To this end, we measured pupillary responses to pictures of smiling faces of strangers (on average rated by the participants as not socially relevant and not familiar), celebrities (non-relevant but familiar) and experimenters (relevant and familiar). We hypothesised that we would observe increased pupil responses in response (1) to more familiar and socially relevant faces and (2) in the repeat-a-pattern game rather than in the passive viewing task. In the passive viewing task, none of the hypothesised predictors showed significant effects, while in the game we unexpectedly observed decreased pupil sizes as a response to more familiar faces. Additionally, the results showed contrasting tonic changes in pupil size in the two tasks: with time (subsequent trials) the pupil size became smaller in the passive viewing task and larger in the game. Reaction times in the game were not modulated by any of the variables of interest. Finally, in an exploratory analysis, we investigated the effects of autistic traits and their interactions with familiarity and social relevance in neurotypical participants in our tasks. As anticipated, we observed that higher autistic traits were linked to smaller pupil sizes when viewing less familiar faces and to larger pupil sizes for more familiar faces.

### 4.1. Subjective Ratings of the Familiarity and Social Relevance of the Stimuli

To ensure that the values of familiarity and social relevance in our study are widespread, we created a stimulus set comprising of three basic face types: strangers, celebrities, and experimenters. Strangers are socially non-relevant to the participants, as they share no social context, and unfamiliar, as they have never met. In contrast, experimenters become familiar and socially relevant to the participants in the context of the study and due to the socialising procedure implemented in its design. Finally, the celebrities, although not personally acquainted or relevant, are characterised by a certain familiarity due to media exposure. We collected participants’ subjective ratings of social relevance and familiarity of the stimuli (as well as attractiveness and reward value in both tasks). Subjective ratings are a popular method of reflecting qualitative values of stimuli, and are also used for familiarity [59,60]. The ratings showed that, on average, the experimenters were indeed perceived as more socially relevant and familiar than the other two faces, and celebrities were also rated as more familiar than strangers. Although the distribution of the ratings showed a considerable intraindividual variability, our results reflected this variance by including subjective ratings as predictors in the planned analyses (for familiarity and social relevance ratings, including raw data points and means, see Appendix A).

### 4.2. Social Reward and Pupillary Responses

#### 4.2.1. Social Relevance

We hypothesised that faces of high social relevance and familiarity would be linked to larger pupil sizes in both tasks, especially in the repeat-a-pattern game. Against our predictions, we observed no effects of social relevance in either of the tasks. One reason for this may be that the experimenters, although significantly more relevant to the participants than celebrities and strangers (as suggested by the subjective ratings), were not relevant enough for the pupillary responses to reflect this effect. Indeed, social relevance has so far been discussed in the context of individuals linked to strong emotional responses, i.e., to loved ones [15]. Hence, the experimenters might not have been relevant enough to elicit a strong reward-related response. However, since to our knowledge no study has so far linked pupil behaviour to social relevance of faces (and specifically study experimenters), our observation is novel and requires more research to shed light on this interpretation.

Alternatively, the ratings of social relevance and familiarity in our models may in fact have explained similar parts of the overall variance and, by including both terms in the regression analysis, the true effect of social relevance might have been occluded. In such a case, a model without familiarity could reveal a larger estimate of social relevance and should be penalised less for overfitting. This, however, was not the case (for this exploratory testing, see points 4.2.2 and 5.2.2 in the analysis code, https://osf.io/623jg/ (accessed on 16 June 2021)).

Finally, it is possible that social relevance of a face is simply not a good predictor of its reward value (at least without engagement of strong emotions as is the case for loved ones). Indeed, although in the repeat-a-pattern game the subjective reward value showed correlations with both social relevance and familiarity, this relationship was stronger for the latter. Additionally, self-rated reward value in the passive viewing task did not correlate with social relevance, even though it did with familiarity. Nonetheless, this result should be viewed with caution, as reward value may not be a meaningful concept in the context of positive stimuli presented in a passive viewing task. This is in line with the otherwise surprising higher ‘reward’ ratings for the stranger than other faces in this task. Overall, these results suggest that social relevance (at least in the range present in this study) of smiling faces does not play a significant role for the reward value of such stimuli.

#### 4.2.2. Familiarity

Although our analyses revealed an effect of familiarity on pupillary responses, it was only manifested in the repeat-a-pattern game and, in contrast to our hypothesis, showed a negative direction: more familiar faces were linked to smaller pupil sizes. Importantly, previous research has provided similar findings. For example, Schneider et al. [30] observed greater and longer dilation for no-reward outcomes than for both monetary and non-monetary outcomes (reported in the Appendix A). If, as indicated in the subjective ratings in our study, the experimenters and celebrities were indeed more rewarding than strangers for the participants in the repeat-a-pattern game, our pattern of results (smaller pupils in response to more rewarding outcomes) parallels those of Schneider and colleagues. This, however, stands at odds with previous studies reporting increase in pupil sizes in response to higher magnitudes of rewards [29] and addiction-related images [32]. Notably, in these studies the pupil behaviour was measured in response to an incentive and not to an actually received reward. Given that the neural and behavioural differences between reward anticipation and reception are well established [61], it is possible that the pupil-coded reward reception targeted in our study does not follow the same pattern of responses as in the studies investigating anticipation.

Since neither the subjective ratings of social relevance nor the reward value predicted pupillary responses, our data did not yield evidence for a straightforward relationship between pupil size and reward. Nonetheless, we consider a few interpretations for the role of pupillary responses in social reward processing. One possible explanation for the observed results is that pupil sizes do not actually reflect reward value of a stimulus via arousal, but rather mere physiological arousal (not modulated robustly by the stimulus). Although we did not directly ask participants to rate their arousal in response to the stimuli, pupil sizes have been repeatedly used as its proxy [27]. However, this selective interpretation stands in contrast to the growing body of literature utilising pupillometry in reward research (e.g., [28,29,30,31,32]). Moreover, if the negative relationship of pupil sizes and familiarity in our study was indeed a reflection of reduced arousal to more familiar faces, this contradicts previous reports of familiar faces (especially with happy expressions) judged as more arousing than unknown ones [36].

Alternatively, the pupil may reflect a reward-related mechanism, however, instead of coding the value of a reward (or from a subjective point of view: appreciation of a reward), it rather reflects its motivational power. Indeed, it has recently been proposed that pupils reflect the activity of the LC neurons, which play a significant role in mobilising energy and resources necessary to perform future actions [62]. Indeed, in some studies the pupil is found only to be modulated by reward magnitude in difficult tasks, in which recruitment of resources and effort are needed to perform them [63]. This interpretation is further supported by the fact that the effect of familiarity was observed in this study in the repeat-a-pattern game but not in the passive viewing task. While in the repeat-a-pattern game, after receiving feedback, the participants were preparing to perform again in the next trial, in the passive viewing task no action was required. Under this interpretation, our data suggest that receiving positive feedback from unfamiliar persons mobilises more resources to perform than does feedback from known faces. A tentative reason for this may lie in the desire to perform well in front of persons about whom we cannot be sure whether they have positive or negative attitudes towards us (such as strangers), in contrast to more familiar experimenters, who are known to be pleasant and helpful. While not disputing that this is a possible interpretation of the obtained results, we recognise that our paradigm does not provide a robust support for it and that different designs are needed to specifically tackle this question.

Finally, our results may reflect a link between pupil size and surprise. In this view, the pupil does not reflect expected reward or uncertainty per se, but rather errors in judging uncertainty, i.e., surprise [64]. Although in our paradigm the different face types were displayed in a random order, it is possible that receiving a smile from an unknown person as feedback for one’s actions is surprising, as it rarely happens in natural situations. Moreover, this interpretation explains the lack of effects in the passive viewing task: as no performance-dependent feedback was included in this task, no surprise could arise, and thus none of the faces were perceived as being more surprising than others.

### 4.3. Reward Contingent on Behaviour

In this study, we contrasted two tasks: a repeat-a-pattern game, in which participants were asked to quickly repeat a pattern and then received feedback dependent on their performance, and a passive viewing task, in which no action was required, and participants were only asked to pay attention to the stimuli. The aim was to compare the reward value of positive stimuli serving as feedback (rewards) and positive stimuli presented regardless of one’s actions. Importantly, the stimuli (smiling faces) in both tasks were the same for each participant. Thus, the difference in pupil responses to the stimuli between the tasks were essentially due to their contingency on performance (in the repeat-a-pattern game) or lack thereof (in the passive viewing task). We hypothesised that the modulatory effects of the stimuli (i.e., social relevance and familiarity) would be larger in the game. Indeed, familiarity of the smiling faces was a significant predictor of the pupil size in the game, but not in the passive viewing task. We believe that the reason for this is that positive stimuli can be considered to be rewards only when their delivery is contingent on one’s behaviour.

Although the ratings of reward value in the passive viewing task should be treated with caution for the abovementioned reasons, a pattern of results emerging from the subjective ratings supports that which is found in the autonomic responses. The reported reward values of the smiling faces were higher in the game and the reward values of the faces correlated positively with social relevance and attractiveness in the game, but not in the passive viewing task. Overall, this suggests that the feedback context (i.e., contingency on behaviour) changes the way social rewards are processed in contrast to passively viewed positive stimuli not contingent on one’s behaviour. Additionally, it is worth mentioning that the two tasks in this study were linked to contrasting effects of time. Specifically, in the game the pupil sizes increased over trials and in the passive viewing task, they decreased. The likely reason for these effects is tonic changes in the tasks: the passive viewing task was monotonous and the decreased pupil size over time could reflect growing fatigue, whereas the repeat-a-pattern game was entertaining and the motivation for performance retained with increasing speed of the pattern presentation over blocks. This possibly led to increased arousal. These effects once again point to the differential engagement in the tasks, and through that with the rewards.

### 4.4. Exploratory Analyses

#### 4.4.1. Autistic Traits and Pupillary Responses

In addition to the main hypotheses, we explored the effect of autistic traits on the pupillary responses to smiling faces in both tasks. Autistic traits were not a good predictor of the pupil size in either of the tasks. However, we observed a predicted descriptive interaction of autistic traits and familiarity of the stimuli in the passive viewing task. The higher the autistic traits, the smaller the pupil responses to less familiar faces. This result parallels that of Nuske et al. [33], who reported reduced pupillary responses to fearful expressions of unfamiliar people (but not familiar ones) in children with ASC relative to typically developing controls. Our exploratory analysis furthers that finding by pointing towards a descriptive effect in a sample of neurotypical subjects differing in the levels of autistic traits. Such results are interesting because they show that social difficulties characterising ASC are likely mediated by the familiarity of others.

#### 4.4.2. Correlations of Attractiveness and Reward Value

It has been suggested that attractive faces can be processed similarly to rewards, and indeed, activation of the reward circuitry in the brain in response to beautiful faces has been previously reported [42]. A positive correlation between subjective ratings of familiarity and reward values in our tasks further supports this claim. However, we only observed such a correlation in the repeat-a-pattern game and not in the passive viewing task. This suggests that, at least on the subjective level, attractive faces are indeed perceived as more rewarding when they serve as feedback contingent on behaviour. This correlation, however, should be viewed with a certain degree of caution, as it is based on a relatively small amount of data, and it does not inform about the causality of the relationship between attractiveness and reward.

### 4.5. Limitations

This study is not free of limitations. A growing body of literature emphasises the need to increase the ecological validity in experiments by including dynamic instead of static faces as stimuli [65]. Here, we used static stimuli to ensure higher control over their physical properties, mainly luminance, which is a crucial factor in pupillometry. Nonetheless, to make the social stimuli more naturalistic, dynamic counterparts of pictures of smiling faces should be employed in future studies.

Particular to this study, it should be noted that we only included female faces in the stimuli set. This was done to match the physical gender of the experimenters (two females). However, a replication of our results with a multi-gender stimulus set is needed before they can be generalised further.

Thirdly, it should be noted that all variables of interest in our models explained a small fraction of the total variance (and thus showed small effect sizes). However, effect sizes in social psychology are commonly smaller than the traditional thresholds for ‘small’, ‘medium’, and ‘large’ effects would suggest [66]. Moreover, cognitive factors typically explain a strikingly small variance in pupil size in comparison to physiological changes such as blinks or tonic fluctuations [67]. Nevertheless, it would be of great benefit to compare our results to similar paradigms employing other psychophysiological indicators of reward processing (e.g., event-related brain potentials).

Fourthly, the subjective ratings of familiarity and social relevance (as well as attractiveness and reward value) were taken at the end of the study, which grasps the final subjective impressions of the participants. Hence, in the unlikely case that the levels of subjective familiarity and relevance changed dynamically throughout the time of the experiment, this might not be reflected in the data.

Finally, despite a growing number of published works exploring pupil behaviour as an indicator of reward processing, this is still a relatively narrow and largely unexplored field. While on the one hand it is difficult to propose convincing interpretations of the results obtained, on the other hand our results emphasise the need to invest more efforts in this research path.

## 5. Conclusions

This study set out to explore the modulatory effects of familiarity and social relevance of social rewards on pupillary responses in tasks in which the reception of a reward was or was not dependent on one’s performance. It provides two major insights. Firstly, familiarity plays a role in the processing of social rewards. Known and unknown faces, regardless of their social relevance, influence the physiological responses to rewarding outcomes differently. Secondly, feedback context is crucial in reward research as positive stimuli are (more) rewarding when they are contingent on behaviour. Both the psychophysiological measurements (pupil dilations) and behavioural responses (subjective ratings) suggested that the feedback context substantially changes how the rewards are processed. The pupil sizes were modulated by familiarity of the rewarding faces only when these faces followed a successful performance. Overall, the findings of this study contribute to our understanding of the social reward processing by targeting the crucial components of the human socio-cognitive functioning: familiarity and social relevance.

## Figures and Tables

**Figure 1 brainsci-11-00794-f001:**
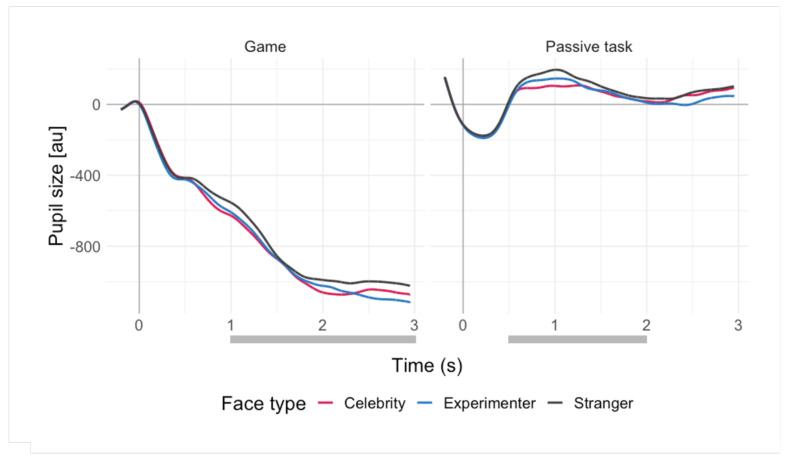
Average pupil sizes (in arbitrary units) in response to stimuli in both tasks across face types (celebrity, experimenter, stranger). The averages are aligned at time 0, which marks the onset of a stimulus presentation (the 200 ms prior to this point comprise the baseline). Grey lines mark the time window used in analyses. Note that the responses are grouped for face types for the purposes of visualisation only (the analyses were conducted with subjective ratings of familiarity and social relevance).

**Figure 2 brainsci-11-00794-f002:**
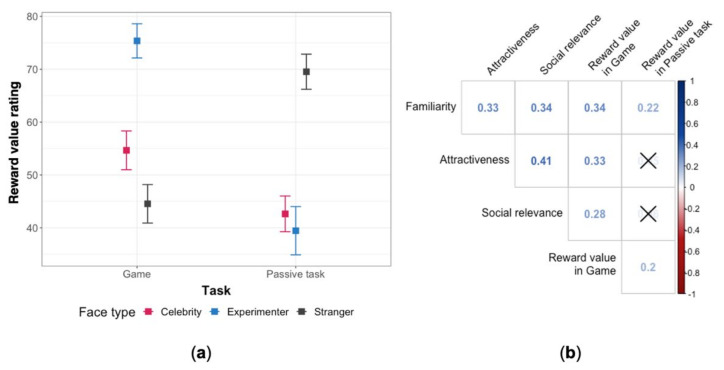
(**a**) Mean subjective ratings of the reward value of each type of face in the repeat-a-pattern game and the passive task. The error bars represent standard errors. (**b**) Correlogram (Pearson) of the ratings: social relevance, familiarity, reward value, and attractiveness. For clarity, the crosses mark statistically insignificant correlations (*p*-values were adjusted with Holm correction), however, this is an exploratory analysis and *p*-values should not be considered relevant.

**Figure 3 brainsci-11-00794-f003:**
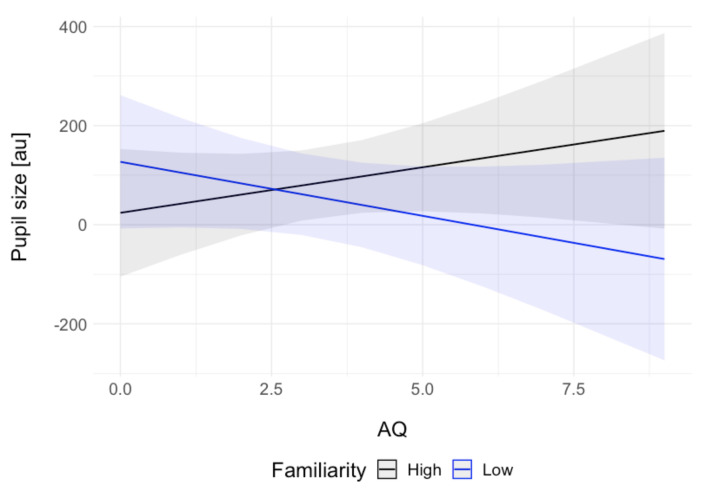
Interaction effects of autistic traits (AQ) and familiarity on pupil size in the passive viewing task. For the purpose of visualisation, the continuous familiarity rating was divided into two factors at value 50 (the range was 1 to 101) into low and high familiarity. The grey areas show 95% confidence intervals.

**Table 1 brainsci-11-00794-t001:** Mean subjective ratings (and standard deviation) of social relevance, familiarity, attractiveness, and reward value for each type of face (experimenter, celebrity, stranger). *p*-values < 0.01, <0.001 are marked with **, and ***, respectively.

	Social Relevance	Familiarity	Attractiveness	Reward Value	
				Game	Passive
Experimenter	42.77 (30.34)	69.53 (27.84)	57.01 (23.84)	75.37 (27.03)	39.46 (38.18)
Celebrity	26.51 (31.86)	75.37 (27.03)	71.76 (23.78)	54.66 (30.68)	42.64 (28.32)
Stranger	11.16 (19.55)	39.46 (38.18)	47.91 (29.71)	44.54 (30.45)	69.53 (27.84)
ANOVA					
Face type	*F*(2,140) = 47.42 ***	*F*(2,140) = 50.07 ***	*F*(2,140) = 17.76 ***	*F*(2,350) = 5.27 **	
Task	-	-	-	*F*(1,350) = 9.41 **	
Face type × Task	-	-	-	*F*(2,350) = 50.51 ***	

**Table 2 brainsci-11-00794-t002:** Mixed effects models investigating the effects of social relevance, familiarity, and reward value on pupil size in the repeat-a-pattern game and in the passive viewing task (with attractiveness and trial number as covariates).

Estimates (95% CI)	Passive Viewing Task		Repeat-a-Pattern Game	
	Estimates	*t*-Value	Estimates	*t*-Value
Intercept	174.00 *** (99.02–248.97)	4.55	−993.36 *** (−1155.60–−831.12)	−12.00
Social relevance	−0.08 (−1.21–1.06)	−0.13	0.83 (−0.62–2.28)	1.12
Familiarity	0.43 (−0.58–1.45)	0.84	−1.72 * (−3.03–−0.41)	−2.57
Attractiveness	−0.08 (−1.26–1.10)	−0.13	1.41 (−0.10–2.92)	1.83
Trial number	−2.15 *** (−2.98–−1.32)	−5.06	0.96 * (0.02–1.89)	2.01
Game: Reward value			−0.71 (−1.99–0.57)	−1.09
**Random Effects**				
σ^2^	756,522.55		1,394,768.97	
τ_00_	41,903.35_code_		415,711.60_code_	
	3206.80_item_			
ICC	0.06		0.23	
N	10_item_		70_code_	
	70_code_			
Observations	6211		6408	
Marginal R^2^/Conditional R^2^	0.004/0.060		0.003/0.232	

* *p* < 0.05, *** *p* < 0.001, ICC = interclass correlation coefficient, τ_00_ = between-subject-variance.

**Table 3 brainsci-11-00794-t003:** Exploratory mixed effects models investigating the effects of autistic traits (AQ score) and their interactions with social relevance and familiarity in the game and in the passive viewing task.

Estimates (95% CI)	Passive Viewing Task			Repeat-a-Pattern Game		
	Estimates	Std. Error	*t*-Value	Estimates	Std. Error	*t*-Value
Intercept	172.44 (96.13–248.74)	38.93	4.43	−992.09 (−1154.76–−829.41)	83.00	−11.95
Social relevance	−0.10 (−1.25–1.06)	0.59	−0.16	0.78 (−0.69–2.25)	0.75	1.04
AQ	2.73 (−25.94–31.39)	14.63	0.19	−17.11 (−99.74–65.53)	42.16	−0.41
Familiarity	0.33 (−0.71–1.36)	0.53	0.62	−1.59 (−2.92–−0.27)	0.68	−2.36
Attractiveness	0.05 (−1.15–1.25)	0.61	0.08	1.36 (−0.16–2.87)	0.77	1.76
Trial number	−2.15 (−2.98–−1.31)	0.42	−5.06	0.95 (0.02–1.89)	0.48	2.00
AQ × Social relevance	0.02 (−0.56–0.61)	0.30	0.07	0.07 (−0.76–0.90)	0.42	0.17
AQ × Familiarity	0.43 (−0.02–0.88)	0.23	1.86	−0.47 (−1.12–0.18)	0.33	−1.43
Game: Reward value				−0.58 (−1.87–0.70)	0.66	−0.89
**Random Effects**						
σ^2^	755,831.57			1,394,162.10		
τ_00_	42,239.43_code_			418,152.94_code_		
	3641.95_item_					
ICC	0.06			0.23		
N	70_code_			70_code_		
	10_item_					
Observations	6211			6408		
Marginal R^2^/Conditional R^2^	0.005/0.062			0.004/0.234		

*p*-values are intentionally not provided due to the exploratory nature of the model. ICC = interclass correlation coefficient, τ_00_ = between-subject-variance.

## Data Availability

The data and analysis code in R for the current study (as well as an html file presenting all the analyses in an accessible way without the need to run the code) are available in the OSF repository: https://osf.io/623jg/ (accessed on 16 June 2021).

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
