# Peer review of "Pupillary Responses to Faces Are Modulated by Familiarity and Rewarding Context"

_brainsci, 2021, doi:10.3390/brainsci11060794_

Round 1

Reviewer 1 Report

The authors evaluated pupil responses to faces in reward and familiarity contexts. The findings are interesting and the manuscript is well-written. Two minor comments about figure 2: 1. gray bars indicate analysis windows. How did the authors decide their analysis windows which are different between panels. 2. Did the authors normalize their data? If so, how and where is the baseline data?

Author Response

We thank the Reviewer for the positive evaluation and the questions about Figure 2, to which we reply below.

The data for each participant were standardised at (near) zero at the onset of stimuli presentations with a 200 ms subtractive baseline correction defined as -200 to 0 ms pre-stimulus onset. This is a standard procedure (and more robust than divisive baseline correction; Mathôt et al., 2018), which allows interpreting changes in pupil size instead of absolute values. The baseline is plotted together with the in-trial pupil responses in Fig. 2. Following the Reviewer’s comment, to make it clearer, we have added to the figure’s legend in the revised manuscript that “The averages are aligned at time 0, which marks the onset of a stimulus presentation (the 200 ms prior to this point consist the baseline)”. Further, the analysis windows (indicated in Figure 2 with grey horizontal bars) were chosen based on the grand averages of the responses in each task. Averaging pupil sizes over arbitrary time windows set for specific times is a common approach in pupil research (for examples, see Geangu et al., 2011; Honma et al., 2012; Sirois & Brisson, 2014). As pre-registered, we planned to exclude from the analysis window up to 1s directly after stimulus onset as a way of removing light response from our models. Since the tasks differ, it is not surprising that the grand averages also showed varying latencies of the initial constriction. This resulted in removing the first 1 and 0.5 s in the active and passing task, respectively. Finally, in the passive task we decided to use the analysis window lasting to 2s, as at that time the pupil averages returns to baseline (whereas the response pattern in the active task continues until the end of the available time window in the trial). Please note that in the analyses we do not directly compare the pupil responses between the tasks. Rather, we estimate the role that the predictors of interest (familiarity, social relevance; and covariates: attractiveness, trial number) play in each of the tasks. Given that and the fact that, as we argue, a reward contingent on behaviour is not identical to a passively viewed positive stimulus, we believe that varying time analysis windows in the two tasks are appropriate.

Cited literature:

Geangu, E., Hauf, P., Bhardwaj, R., & Bentz, W. (2011). Infant Pupil Diameter Changes in Response to Others’ Positive and Negative Emotions. PLoS ONE, 6(11), e27132. https://doi.org/10.1371/journal.pone.0027132

Honma, M., Tanaka, Y., Osada, Y., & Kuriyama, K. (2012). Perceptual and not physical eye contact elicits pupillary dilation. Biological Psychology, 89(1), 112–116. https://doi.org/10.1016/j.biopsycho.2011.09.015

Mathôt, S., Fabius, J., Van Heusden, E., & Van der Stigchel, S. (2018). Safe and sensible preprocessing and baseline correction of pupil-size data. Behavior Research Methods, 50, 94–106. https://doi.org/10.3758/s13428-017-1007-2

Sirois, S., & Brisson, J. (2014). Pupillometry. WIREs Cogn Sci, 5, 679–692. https://doi.org/10.1002/wcs.1323

Reviewer 2 Report

The authors have done an extremely thorough job in this manuscript. The introduction provides a well-reasoned rationale for the study, the methods are straightforward, the results appropriately in depth, and the discussion does not overstate the findings. The findings themselves deviate from predictions, in several statistically significant ways. Perhaps the most surprising aspect is that the relationship between pupil dilation and familiarity is opposite to what was predicted, namely that it should increase rather than decrease with familiarity. For as far as possible accounts in terms of whether the stimuli were at all rewarding is concerned, this is a reasonable concern but the authors address is extensively. Altogether, I feel that the manuscript is close to publication.

The only minor comments I have are the following two points:

  1. The shadings of gray in figure 3 are not clear. They look much too similar.
  2. The manuscript could benefit from rewrites in some sections. In particular, sentences run on for too long at times. It would help readability if multiple aspects of an idea were to be expressed in separate sentences. At times, one sentence seems to go on for 5 or more lines of text.

Author Response

We thank the Reviewer for the positive evaluation of our manuscript and for the comment on Figure 3. The grey shadings indicate the 95% confidence intervals (as stated in the figure’s legend), but we agree that the same grey shade makes it hard to differentiate them. In the corrected manuscript Figure 3 includes two colours: black and blue for clarity.

We are also grateful for the second point raised by the Reviewer, relating to the length of some of the sentences in the manuscript. We reviewed it with a special attention to this aspect and we edited such phrases (especially in the Introduction) making sure that they are either realised in two or more sentences, or the reading is facilitated with clearly marked structure (e.g., numbered points dividing a sentence according to the structure of an argument). Moreover, we have conducted a professional English proofreading of the manuscript (by the Kelly GmbH), which further improved the clarity of the text. These changes are accepted in the revised manuscript. We believe that thanks to this, the manuscript has improved.